# Capturing Structural Variants of Herpes Simplex Virus Genome in Full Length by Oxford Nanopore Sequencing

Rajagopalan Saranathan,[a] Emmanuel Asare,[a] Lawrence Leung,[a] Anna Paula de Oliveira,[a] Katherine E. Kaugars,[a] Claire V. Mulholland,[a] Regy Lukose,[a] Michael Berney,[a] ⓘWilliam R. Jacobs, Jr.[a]

[a]Department of Microbiology and Immunology, Albert Einstein College of Medicine, New York City, New York, USA

**ABSTRACT** Genome sequencing and assembly of viral genomes within the *Herpesviridae* family, particularly herpes simplex virus (HSV), have been challenging due to the large size (~154 Kb), high GC content (68%), and nucleotide variations arising during replication. Oxford Nanopore Technology (ONT) has been successful in obtaining read lengths ranging from 100 Kb up to 2.3 Mb. We have optimized DNA extraction and sequencing with ONT to capture the whole genome of HSV-1 as a single read. Although previous studies described the presence of four different genome isomers of HSV, we provided the first report on capturing all four variants' full-length genome as single reads. These isomers were found to be present in almost equal proportion in the sequenced DNA preparation.

**IMPORTANCE** With the advent of next-generation sequencing platforms, genome sequencing of viruses can be performed in a relatively shorter time frame in even the most austere conditions. Ultralong read sequencing platforms, such as Oxford Nanopore Technology (ONT), have made it possible to capture the full-length genome of DNA viruses as a single read. By optimizing ONT for this purpose, we captured the genome (~154 Kb) of a clinical strain of herpes simplex virus 1 (HSV-1). Additionally, we captured full-length sequences of the four isomers of lab-grown HSV-1 virus and were able to determine the frequency of each within the isogenic population. This method will open new directions in studying the significance of these isomers and their clinical relevance to HSV-1 infections. It will also improve basic studies on the recombination and replication of this virus.

**KEYWORDS** genome isomers, Oxford Nanopore sequencing, herpes simplex virus

Address correspondence to William R. Jacobs, Jr., william.jacobs@einsteinmed.edu.

WRJ is a co-inventor of Recombinant Herpes Vaccine Vector technologies, which was developed in the laboratory of WRJ and Betsy Herold, is described in two U.S. patent applications (10,980,874 and 9,999,665), and is licensed to X-Vax Technology, Inc. WRJ is a consultant and stockholder of X-Vax Technology, Inc., and WRJ receives financial support from X-Vax Technology, Inc. for research.

*[This article was published on 30 August 2022 without a conflict of interest statement. The statement was added to the revised version, posted on 8 September 2022.]*

Herpes simplex virus 1 (HSV-1) is a large double-stranded DNA virus that belongs to the *Alphaherpesvirinae* subfamily within the *Herpesviridae* family. HSV-1 is widespread in the human population, causing mild to severe skin and neurological infections (1–3). The genome of HSV-1 is complex and comprised of unique long ($U_L$) and unique short ($U_S$) regions flanked by repeat elements on both ends. The termini of unique long ($U_L$) and short ($U_S$) regions are flanked by the terminal and internal repeat of the long region (TRL and IRL) and the terminal and internal repeat of the short region (IRS and TRS) (Fig. 1A) (4, 5). Although HSV-1 has been reported to form genome isomers because of recombination events during replication, the significance of these isomers in virulence or viral fitness is not completely understood (6). The presence of these isomers was identified earlier by restriction digestion, southern blotting, and probe-based methods (7–11). Capturing HSV variants with their nucleotide sequence has been a challenge using short-read shotgun sequencing platforms due to extremely high GC content (70 to 85%) in some regions, such as the inverted repeats of the genome; genetic variations; and complexities associated with isolating intact viral genomic DNA (12–16). In addition, short reads would need extensive coverage and computational processing and thus, cannot be reliably assembled to generate a full-length HSV-1 genome. Recently, Chang et al. (17) identified the DNA isomers of HSV-2 strain G by ONT but reported them as partial genome fragments. Because some of the replication- and virulence-related genes are present in repeat regions, capturing both copies of repeat elements is important to

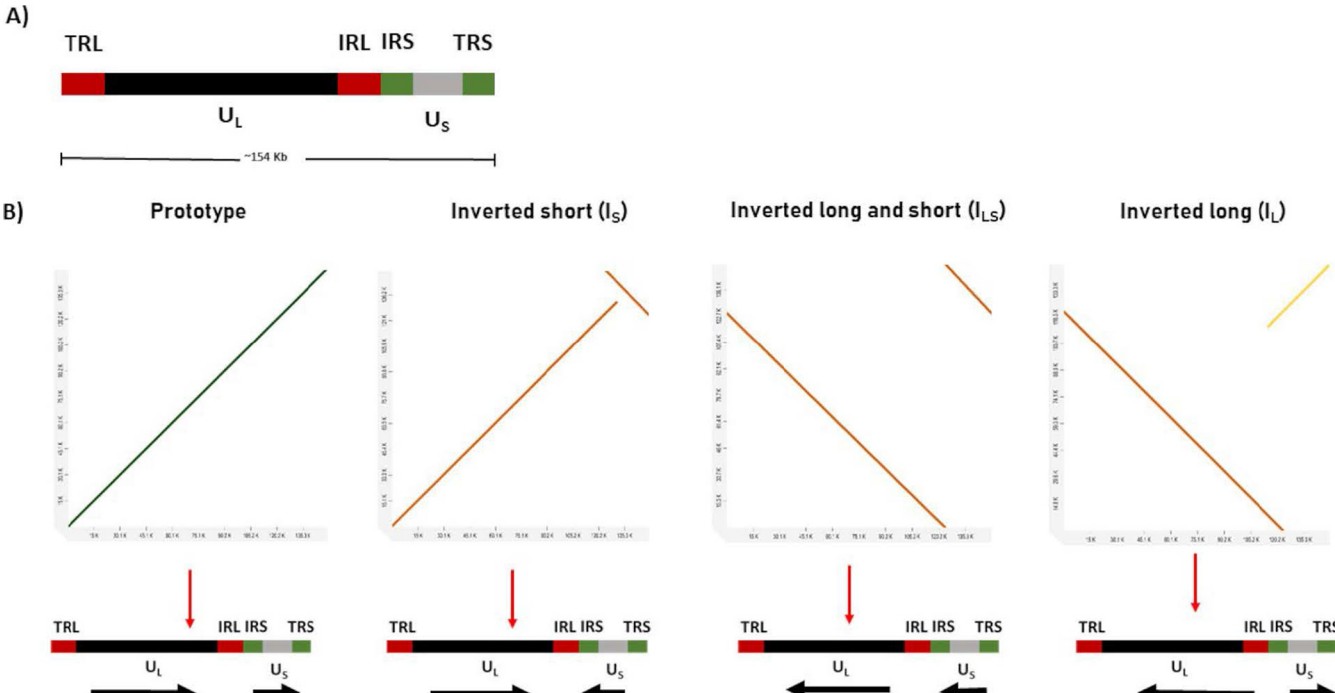

**FIG 1** HSV-1 B³x 1.1 genome variants identified by Nanopore sequencing. (A) Schematic representation of HSV-1 genome with the repeat elements (TRL, IRL, TRS, IRS) on both ends of unique long ($U_L$) and unique short ($U_S$) regions. (B) All four isomers, such as prototype (P), inverted short ($I_S$), inverted long and short ($I_{LS}$), and inverted long ($I_L$) were identified by performing dot plot. Ultralong reads ranging from 145 kb to 155 kb were mapped against a reference genome (accession no. NC_001806) to identify the genome isomers using http://dgenies.toulouse.inra.fr/run.

understand the process, but this is possible only with the full-length genome sequence (8). Furthermore, obtaining a full-length accurate HSV genome will be valuable for studying determinants of drug resistance, virulence, pathogenesis, and viral evolution (16). In this observation, we performed a hybrid assembly to obtain the genome sequence of an HSV-1 clinical strain using both ONT long reads and Illumina short reads.

We recently developed a method to extract intact full-length genomic DNA from HSV-1 and HSV-2, followed by sequencing and capturing their genome as a single read by ONT. Genomic DNA from HSV-1 clinical strain (B³x 1.1) (18) was isolated following the protocol optimized earlier (19). Briefly, Vero cells were infected at a multiplicity of infection (MOI) of $10^{-2}$ PFU/cell for 1 h at 37℃. When 90 to 100% cytopathic effect was observed, the supernatant and cells were collected and centrifuged at 300 $\times$ $g$ for 10 min; the cell pellet was washed and resuspended in 10 mM Tris (pH 7.5) with protease inhibitor cocktail, homogenized, and centrifuged at 3000 $\times$ $g$. The pellet was resuspended in 10 mM Tris (pH 7.5) containing 5 mM $MgCl_2$ and 180 units of benzonase and incubated for 30 min at room temperature. This was followed by centrifugation. The pellet was resuspended in TNE (50 mM Tris pH 7.5, 0.1 M NaCl, 10 mM EDTA), centrifuged 10,000 $\times$ $g$, followed by resuspension in TNE containing 0.5% SDS and 0.5 mg/mL pronase and incubated for 1 h at 37℃. The sample was then extracted with phenol:chloroform:isoamyl alcohol (25:24:1). The DNA was centrifuged at 15,000 rpm, washed with 70% ethanol, resuspended in a precipitation solution (5 M NaCl and 13% PEG 8000 in distilled water), and placed on ice. The sample was then centrifuged at 15,000 rpm, washed with 70% ethanol, and dissolved in TE buffer. The genomic library was prepared with the extracted DNA using the Ligation Sequencing kit (SQK-LSK109) following the manufacturer's instructions and sequenced using a MinION sequencing unit. Wide orifice pipette tips were used throughout the extraction and library preparation procedure. We also performed shotgun sequencing on B³x 1.1 genomic DNA using a benchtop Miseq sequencer (Illumina, USA). The genomic library was prepared using the Nextera XT library preparation kit and sequenced with a 150-cycle (2 $\times$ 75 bp) v3 reagent kit following the manufacturer's instructions.

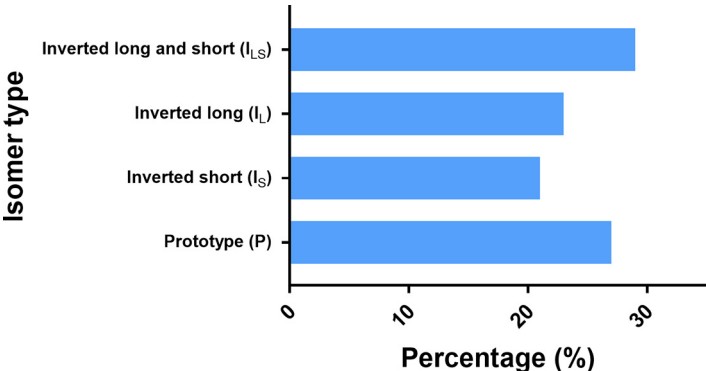

**FIG 2** Quantification of HSV-1 B³x 1.1 genome isomers. Quantification of HSV-1 B³x 1.1 genome isomers was performed using Geneious prime (version 2022.0.1). Reads that were more than 17 kb were retained and among which the sequences with both inverted repeat sequence elements (IRL and IRS) with at least 1 kb sequence on both ends were selected. Among the selected quantifiable reads prototype (P), inverted long ($I_L$), inverted short ($I_S$), and inverted long and short ($I_{LS}$) isomers were found to be 27%, 21%, 23%, and 29%, respectively.

With ONT sequencing, we obtained 104,524 reads with an average length of 1294 bp. We also obtained 15 ultralong reads >145 Kb, among which six reads were >150 Kb in length, and captured the whole genome of the virus. To identify the genome isomers of the virus and their proportion, we performed dot plot analysis using the HSV-1 prototype (accession no. KU310657.1) as a reference genome. We observed four different forms of plots with different sequence files signifying the presence of all four isomers in our DNA preparation (Fig. 1B). We also annotated the 15 ultralong reads of which six were prototype (P), five were inverted long type ($I_L$), one was inverted short type ($I_S$) and three were inverted long and short type ($I_{LS}$). Due to the relatively high error rate of Oxford Nanopore sequencing, we used Illumina MiSeq data to polish ONT genomes for each of the four isomers. Around 3 million short reads (75 bp) were obtained providing a mean mapped read depth of ~900 to 1000×. Following trimming with Trim Galore we used BWA-MEM (https://arxiv.org/abs/1303.3997) to map short reads to ONT sequences and corrected base errors (SNPs, indels, and ambiguous calls) using Pilon v1.24 (20). We performed four iterative cycles of genome polishing with BWA-MEM and Pilon after which no further gains were observed.

We quantified the percentage of isomers from the ONT sequence data by retaining reads which are >17 Kb in length using Geneious Prime software. The filtered sequences were annotated and segregated by picking sequences that had both inverted repeat sequence elements (IRL and IRS) with at least 1 Kb sequence on both sides to identify the $U_S$ and $U_L$ genes on both sides. We then counted the annotated reads fitting to each isomer and quantified them. Among the 630 sorted quantifiable reads, prototype (P), inverted long ($I_L$), inverted short ($I_S$), and inverted long and short ($I_{LS}$) isomers were found to be 27%, 21%, 23%, and 29%, respectively (Fig. 2). Similar to previous observations on isomer distribution, we detected the isomers to be present in nearly equal percentage in the sequenced genomic DNA preparation.

In conclusion, the manuscript presents an optimized method of DNA extraction and sequencing to capture the whole genome of the HSV-1 virus and its isomers as a single read. Although earlier studies identified these isomers, their significance is not completely understood. One study found that for the HSV-1 KOS strain, the 4 isomers were in equivalent ratios when grown *in vitro*, which is consistent with previous studies (8). However, when the same strain was isolated from the trigeminal ganglia of infected mice, the $I_{LS}$ isomer was predominant, suggesting that some isomers may have a fitness advantage in a living organism (8). The method in the manuscript was optimized to digest host cell DNA and any HSV DNA that is not packaged in virions, leaving only the DNA from encapsulated virions. Therefore, this analysis is likely comprised of DNA from intact HSV virions and would give a more accurate picture of the relative amounts of each isomer in the infective population of viruses.

However, even DNA encapsulated in HSV virus particles may be noninfectious, and the relationship between isomers and infectivity is yet to be fully elucidated. The ultralong read sequencing technologies such as ONT have opened up new directions to study these isomers. In the future, sequencing clinical specimens from patients might provide more useful information about specific isomers and their virulence, or their role in causing infections in humans. Furthermore, this method could also be used to screen mutants of HSV for defects in isomerization, thereby contributing to the understanding of the basic mechanisms of isomer formation.

**Data availability.** Full-length genome of HSV-1_B$^3$x 1.1 isomers B$^3$x 1.1A (P), B$^3$x 1.1B (I$_S$), B$^3$x 1.1C (I$_L$), and B$^3$x 1.1D (I$_{LS}$) were submitted in GenBank and their accession numbers are prototype (ON783214), inverted short (ON783215), inverted long (ON783216) and Inverted Long and short (ON783217). ONT Sequencing data and Illumina paired-end reads were deposited in the SRA database with the accession numbers SRR20796393 and SRR20796392 under the Bioproject ID PRJNA846339.

## SUPPLEMENTAL MATERIAL

Supplemental material is available online only.
**SUPPLEMENTAL FILE 1**, PDF file, 1.7 MB.

## ACKNOWLEDGMENTS

We acknowledge the NIH for our funding from grant R21AI156853-01 and grant R01AI026170-33A1.

WRJ is a co-inventor of Recombinant Herpes Vaccine Vector technologies, which was developed in the laboratory of WRJ and Betsy Herold, is described in two U.S. patent applications (10,980,874 and 9,999,665), and is licensed to X-Vax Technology, Inc. WRJ is a consultant and stockholder of X-Vax Technology, Inc., and WRJ receives financial support from X-Vax Technology, Inc. for research.

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
