## [Reviewer comments · Microbiology Spectrum]

Microbiology Spectrum

Capturing structural variants of herpes simplex virus genome in full length by Oxford Nanopore sequencing

William Jacobs, Saranathan Rajagopalan, Emmanuel Asare, Lawrence Leung, Anna de Oliveira, Katherine Kaugars, Claire Mulholland, Regy Lukose, and Michael Berney

Corresponding Author(s): William Jacobs, Albert Einstein college of Medicine

Review Timeline:

Submission Date:	June 17, 2022
Editorial Decision:	August 3, 2022
Revision Received:	August 5, 2022
Accepted:	August 10, 2022

Editor: JJ Miranda

Reviewer(s): The reviewers have opted to remain anonymous.

Transaction Report:

DOI: <https://doi.org/10.1128/spectrum.02285-22>

August 3, 2022

Dr. William R Jacobs
Albert Einstein college of Medicine
Department of Microbiology and Immunology
1300 Morris park Ave
Price 569
Bronx, New York 10461

Re: Spectrum02285-22 (Capturing structural variants of herpes simplex virus genome in full length by Oxford Nanopore sequencing)

Dear Dr. William R Jacobs:

I thank you for your contribution to those interested in herpesviruses sequencing. Two experts in the field have reviewed this manuscript, and the comments were split on the novelty and significance of the findings. I've also examined the submission and concluded that the work is technically sound and within the scope of Spectrum's mission to publish Observations that are rigorous without consideration of potential impact.

I look forward to a thoughtful revision that includes data sharing depositions and a more thorough review of similar previously published methods.

Link Not Available

Sincerely,

JJ Miranda

Journals Department
Reviewer comments:

Reviewer #1 (Public repository details (Required)):

The raw sequencing data need to be deposited to the Sequencing Read Archive.

Reviewer #1 (Comments for the Author):

This is a very interesting study, the full-genome sequences of the virus are extremely important. The manuscript could gain by citing and commenting on a previous study performed by Karamitros et al. where Nanopore and short read sequenced were combined. <https://www.ncbi.nlm.nih.gov/pmc/articles/PMC4910999/>

Reviewer #2 (Public repository details (Required)):

ONT sequencing data and Illumina sequencing data need to be deposited to NCBI SRA database

Reviewer #2 (Comments for the Author):

In "Capturing structural variants of herpes simplex virus genome in full length by Oxford Nanopore sequencing", authors improved HSV-1 improved viral genome DNA, observed 15 full-length genome of four types isomers and obtained the composition of four isomers based on reads containing IRL, IRS and flanking regions.

Both the viral DNA prep method improvement and virology discovery aren't significant enough for publication.

Staff Comments:

Preparing Revision Guidelines

Please return the manuscript within 60 days; if you cannot complete the modification within this time period, please contact me. If you do not wish to modify the manuscript and prefer to submit it to another journal, please notify me of your decision immediately so that the manuscript may be formally withdrawn from consideration by Microbiology Spectrum.

In "Capturing structural variants of herpes simplex virus genome in full length by Oxford Nanopore sequencing", authors improved HSV-1 improved viral genome DNA, observed 15 full-length genomes of four types of isomers and obtained the composition of four isomers based on reads containing IRL, IRS and flanking regions.

Both the viral DNA prep method improvement and virology discovery aren't significant enough for publication.

William R. Jacobs, Jr., PhD
Professor Microbiology & Immunology
Albert Einstein College of Medicine
william.jacobs@einsteinmed.edu
1301 Morris Park Avenue
Bronx, New York - 10461

Dear Dr. Miranda

We thank you and the two reviewers for taking the time to review our submission and their comments. Please find below the pointwise response in bold letters. The manuscript has been revised and we have highlighted all changes in the manuscript.

Reviewer #1

The raw sequencing data need to be deposited to the Sequencing Read Archive.

We submitted both ONT and Illumina Miseq raw data to NCBI-SRA database and included the SRA number in the revised manuscript

This is a very interesting study, the full-genome sequences of the virus are extremely important. The manuscript could gain by citing and commenting on a previous study performed by Karamitros et al. where Nanopore and short read sequenced were combined.

<https://www.ncbi.nlm.nih.gov/pmc/articles/PMC4910999/>

We thank the reviewer for pointing out this important study. We discussed the findings of this publication and included in the revised manuscript

Reviewer #2

ONT sequencing data and Illumina sequencing data need to be deposited to NCBI SRA database

We submitted both ONT and Illumina Miseq raw data to NCBI-SRA database and included the SRA number in the revised manuscript

In "Capturing structural variants of herpes simplex virus genome in full length by Oxford Nanopore sequencing", authors improved HSV-1 improved viral genome DNA, observed 15 full-length genome of four types isomers and obtained the composition of four isomers based on reads containing IRL, IRS and flanking regions.

Both the viral DNA prep method improvement and virology discovery aren't significant enough for publication.

To our knowledge, this would be first report on capturing full-length sequences of the four isomers as single read from a herpesvirus DNA preparation. This is important as mutations are often found among the genes in inverted repeat elements, which are present in two copies and hard to capture

with shotgun sequencing approaches. Causality of mutation-induced phenotypes could only be concluded with full-length genome sequences. We believe that future work will improve using this DNA isolation method and ONT would be the assay for studying HSV. B³x1.1 is a highly virulent recent clinical isolate that is a valuable strain for future vaccine works and genetic studies.

We thank the reviewers for the opportunity to improve our manuscript,

William R. Jacobs Jr, PhD

August 10, 2022

Dr. William R Jacobs
Albert Einstein college of Medicine
Department of Microbiology and Immunology
1300 Morris park Ave
Price 569
Bronx, New York 10461

Re: Spectrum02285-22R1 (Capturing structural variants of herpes simplex virus genome in full length by Oxford Nanopore sequencing)

Dear Dr. William R Jacobs:

Thank you for your contribution of methods that may further the study of herpesvirus genomics.

Your manuscript has been accepted, and I am forwarding it to the ASM Journals Department for publication. You will be notified when your proofs are ready to be viewed.

Sincerely,

JJ Miranda
Editor, Microbiology Spectrum
